# Effects of Water Availability in the Soil on Tropane Alkaloid Production in Cultivated *Datura stramonium*

**DOI:** 10.3390/metabo9070131

**Published:** 2019-07-03

**Authors:** Abigail Moreno-Pedraza, Jennifer Gabriel, Hendrik Treutler, Robert Winkler, Fredd Vergara

**Affiliations:** 1Department of Biochemistry and Biotechnology, Center for Research and Advanced Studies Irapuato, Guanajuato 36821, Mexico; 2Molecular Interaction Ecology, German Centre for Integrative Biodiversity Research Halle-Jena-Leipzig, 04103 Leipzig, Germany; 3EcoMetEoR, German Centre for Integrative Biodiversity Research Halle-Jena-Leipzig, 04103 Leipzig, Germany; 4Bioinformatics & Mass Spectrometry, Leibniz Institute of Plant Biochemistry, 06120 Halle, Germany

**Keywords:** agriculture, alkaloid, atropine, *Datura*, irrigation, metabolomics, pharmacy, scopolamine, tensiometers, tropane

## Abstract

Background: different Solanaceae and Erythroxylaceae species produce tropane alkaloids. These alkaloids are the starting material in the production of different pharmaceuticals. The commercial demand for tropane alkaloids is covered by extracting them from cultivated plants. *Datura stramonium* is cultivated under greenhouse conditions as a source of tropane alkaloids. Here we investigate the effect of different levels of water availability in the soil on the production of tropane alkaloids by *D*. *stramonium*. Methods: We tested four irrigation levels on the accumulation of tropane alkaloids. We analyzed the profile of tropane alkaloids using an untargeted liquid chromatography/mass spectrometry method. Results: Using a combination of informatics and manual interpretation of mass spectra, we generated several structure hypotheses for signals in *D*. *stramonium* extracts that we assign as putative tropane alkaloids. Quantitation of mass spectrometry signals for our structure hypotheses across different anatomical organs allowed us to identify patterns of tropane alkaloids associated with different levels of irrigation. Furthermore, we identified anatomic partitioning of tropane alkaloid isomers with pharmaceutical applications. Conclusions: Our results show that soil water availability is an effective method for maximizing the production of specific tropane alkaloids for industrial applications.

## 1. Introduction

Tropane alkaloids are bicyclic compounds produced by different genera in the Solanaceae and Erythroxylaceae. These compounds act as anticholinergic agents affecting the central and peripheral nervous system as competitive, non-selective muscarinic acetylcholine receptor antagonists that prevent the binding of the physiological neurotransmitter acetylcholine [1,2]. The tropane alkaloids atropine, scopolamine, and their derivatives are the active ingredients in numerous pharmaceuticals ranging from antidotes for poisoning by organophosphoric compounds, antispasmodics, anti-motion sickness agents, and anti-emetics [3,4,5]. The world health organization includes in its catalog of essential medicines atropine and its derivative ipratropium bromide and the scopolamine derivatives tiotropium bromide and scopolamine-*N*-butyl bromide [6]. Scopolamine-*N*-butyl bromide (Buscopan^®^) and tiotropium bromide (Spiriva^®^) with sales in 2016 of € 236 million and € 2,995 million, respectively, exemplify the lucrative market for tropane alkaloids as pharmaceuticals [7]. Tropane alkaloids can be obtained through organic chemical or biotechnological methods [8,9,10,11,12]. However, these methods remain economically non-competitive in comparison to extraction from plants [13]. Therefore, the demand for tropane alkaloids for industrial applications is covered primarily by plantations. As agriculture remains the primary source for tropane alkaloid production [14] it is of the utmost relevance to understand the effects of environmental factors on tropane alkaloid productivity in outdoors cultures of annual Solanaceae. Water is one of the most limiting factors for agriculture [15,16], and the effects of irrigation on tropane alkaloid production are not fully understood. The Solanaceae *Datura stramonium* is a non-domesticated, annual species that originated in Mexico. *Datura stramonium* has been used by different cultures across North America in folk medicine and shamanism. Atropine and scopolamine are two of the most studied tropane alkaloids produced by *D*. *stramonium*. Wild populations of *D*. *stramonium* have been the focus of research in ecology, particularly in the areas of herbivory, pollination, and population ecology [17,18,19,20]. Cultivated *D*. *stramonium* has been used over a century as a model for plant genetics because of its easy cultivation, short life cycle, the ability for outcrossing and selfing and the existence of several genetically-based, easy to identify phenotypes [21,22]. After the arrival of the Europeans to the Americas in the XV century plants of the genus *Datura* spread throughout the globe, and they are found nowadays as wild populations in other continents [23,24]. Additionally, *D*. *stramonium* is now also cultivated in different countries as a source of tropane alkaloids for pharmaceutical applications.

In this study, we tested the impact of water availability in the soil on the metabolism of tropane alkaloids in *D*. *stramonium*. We recorded water tension in the soil and used it as an indicator of root water uptake. We profiled tropane alkaloids in different organs using mass spectrometry-based, untargeted metabolomics approach. We also determined the absolute concentration of atropine and scopolamine in different organs by a targeted LC-MS-QqQ method. Our results showed that tropane alkaloid fingerprints and absolute concentrations differ in a tissue-specific manner in response to irrigation levels.

## 2. Results

### 2.1. Water Availability in Roots Depending on Irrigation

To evaluate water availability under different irrigation conditions, we measured water tension. The tensiometers we used to measure how tightly water is bound to soil/substrate particles and, as a consequence, the amount of energy the plants need to exert to extract water from a soil substrate. The relation between volumetric water content and matric potential is characteristic of a specific porous medium. The curves generated with the tensiometer readings showed a circadian rhythm in the availability of water for the roots of *Datura stramonium* (Appendix A). All four irrigation levels (500, 1000, 1500, and 2000 mL every 8 days) produced curves with similar sinusoidal shape. The dry weight of the roots and stems reached a maximum at 1500 mL irrigation while leaves with the more considerable dry weight were produced at 1000 mL irrigation (Figure 1).

### 2.2. Profile of Tropane Alkaloids

We applied two exploratory approaches to generate hypotheses for the possible compounds present in the plant extracts; we followed two approaches in parallel.

#### 2.2.1. Metabolite Family Hypotheses

We used the program MetFamily to cluster MS^2^ spectra according to similarities with mass fragmentation patterns. The MetFamily output consisted of 114 features extracted from the MS^2^ spectra. We identified a tropane alkaloid class containing four subtypes: Atropine-like Type I to IV (Figure 2). Besides, we identified a cluster related to caffeic acid and a cluster related to caffeoyl-putrescine. Details about the members within the individual clusters are shown in Appendix A.

#### 2.2.2. Manual Interpretation of Mass Spectra and Relative Signal Quantitation

We generated structure hypotheses for several signals classified as tropane alkaloids by MetFamily by manually interpreting the full scan and MS^2^ spectra produced by LC-MS-qToF. We compared our predictions with spectral data of tropane alkaloids obtained from databases and bibliography. Our structure hypotheses are divided into aromatic and aliphatic tropane esters. Figure 3 and Figure 4 show the relative concentrations of each of our structure predictions in the different irrigations and organs. Appendix A shows the total yield per organ and compound.

### 2.3. Absolute Concentration of Atropine and Scopolamine

We determined the absolute concentration of atropine and scopolamine using a mass spectrometry/multiple reaction monitoring methods. The average concentrations of atropine and scopolamine in the different combinations of irrigations and organs (Appendix A) ranged from 0.02 to 0.1 mg g^−1^ and from 0.02 to 0.4 mg g^−1^, respectively.

### 2.4. Atropine Hydroxylation

As we processed the information in the LC-qToF chromatograms, we recognized isomers of a compound we hypothesize as anisodamine. Figure 5 shows the extracted ion chromatograms for *m*/*z* 306.1 ([M+H]^+^) in different organs of a single plant grown with 1500 mL irrigation. Interestingly, of the two most abundant isomers in roots (retention times of 4.1 and 5.4 min) only the compound eluting later shows similar abundance in roots, stems, and leaves. The concentration of the earlier peak is much lower in stems and leaves.

### 2.5. Elemental Nitrogen Content

Alkaloids are organic compounds containing nitrogen. Consequently, we determined the percentage of nitrogen in the different combinations of irrigations and organs. In the case of the leaves, the percentage of elemental nitrogen decreased with higher irrigations (Figure 6).

## 3. Discussion

### 3.1. Effect of Irrigation on Tropane Alkaloid Accumulation

We show that irrigation can be tuned to maximize the production of specific, medically relevant aromatic tropane alkaloids in *Datura stramonium* cultivated under greenhouse conditions. We also show that, for particular tropane alkaloids, irrigation is more relevant than dry weight to explain compound concentrations. Atropine showed higher concentrations in roots and leaves at 1000 mL irrigation. Anisodamine isomers were more concentrated in roots at 1500 mL irrigation. As for the aliphatic tropane alkaloids, we found that these compounds are more concentrated in roots and that irrigation of 1000 mL increases their concentration. Tropane alkaloids aliphatic esters of the type tigloyloxyl (similar to the predicted structures shown in Figure 4) were described in *Brugmansia sanguinea*, a close relative of *D*. *stramonium* [25,26]. The bioactivity of these compounds remains, to our knowledge, yet to be discovered.

### 3.2. Total Tropane Alkaloid Productivity

The current demand for natural tropane alkaloids as starting materials for pharmaceuticals is covered at a large extent by plantations of *Duboisia* spp. hybrids. Plantations of *Duboisia* have been favored as a source of tropane alkaloids due to their high yields [13]. In this study, concentrations of atropine and scopolamine in all tissue of *D*. *stramonium* were lower as in *Duboisia*. However, as *Duboisia* is a tree, it requires years before reaching optimal productivity [14]. In contrast, the short life cycle of *D*. *stramonium* allows fast, prompt production of tropane alkaloids. The potential for rapid production of tropane alkaloids with *D*. *stramonium* could be relevant in the case of tropane alkaloid shortages caused by sudden diseases affecting *Duboisia* plantations. Fast greenhouse growth conditions allow multiple harvests of *D*. *stramonium* per year and tropane alkaloids can be extracted from whole plants. The information shown in the present study opens the possibility to further optimization of tropane alkaloid productivity based on irrigation. It must be remarked, that tropane alkaloid concentration was not correlated with the total amount of elemental nitrogen.

### 3.3. Aromatic Tropane Alkaloid Metabolism

In vitro assays showed that atropine is enzymatically converted into scopolamine via the intermediate anisodamine [27]. This chemical transformation is conceptualized as two concerted reactions: (i) the hydroxylation of atropine followed by (ii) the cyclization of anisodamine to form an epoxide between the carbon atoms C6 and C7. The two reactions are assumed to be catalyzed in vivo by a single enzyme: hyoscyamine-6β-hydroxylase (H6H, EC 1.14.11.11) [28]. H6H has been cloned from different Solanaceae [29]. When the heterologously expressed enzyme was incubated with atropine, it produced anisodamine and scopolamine but in a ratio of 40:1 [27]. This opens the possibility that another enzyme could perform in vivo the second reaction. Experiments with isotope labelled atropines showed that the H6H introduces a hydroxyl group in a regio-selective manner at carbon C6 of the tropane rings [30]. The experiments were conducted using an H6H cloned from *Hyoscyamus niger* and heterologously expressed in *Escherichia coli* [31]. In addition to the hydroxylation at C6 a small amount of hydroxylation occurred at C7. Assuming that the two main peaks we detected in *D*. *stramonium* roots and whose mass spectra correspond with anisodamine were produced by two independent hydroxylations at positions C6 and C7, the ratio between the regioisomers were approximately equimolar. Consequently, either our *D*. *stramonium* expressed an H6H with no regioselectivity for C6 or our plants expressed two different hyoscyamine-β-hydroxylases. Furthermore, the fact that only the later eluting peak was mostly detected in stems and leaves opens the possibility that either hyoscyamine-β-hydroxylases with specific regioselectivity were expressed in different organs or that out of the two main regioisomers detected in the roots only one of them was transported into other organs. As anisodamine is also used as an anticholinergic agent, like atropine and scopolamine, and because its activity has been related to its stereochemistry the fact that *D*. *stramonium* produces different stereoisomers can indicate specific bioactivities depending on the organ [32,33]. Perhaps natural herbivores of roots and leaves of *D*. *stramonium* respond differentially to specific stereoisomers of tropane alkaloids.

## 4. Materials and Methods

### 4.1. Plant Material and Greenhouse Conditions

The botanic gardens of the University of Leipzig provided us seeds of *Datura stramonium*. The seeds originated from a plant grown and harvested within in the premises of the botanic gardens in the year 2017 (internal registry number XX-0-LZ-APO-39-2010). We germinated the seeds by placing them within a matrix of 1 mm Ø glass beads (Paul Marienfeld GmbH & Co KG). We added plenty of water to keep the seeds always hydrated. After germination, we transferred plantlets into the culture pots equipped with the tensiometers (see Section 4.2). We used a total of 16 plants divided equally into four irrigation levels: 500, 1000, 1500, and 2000 mL. We placed the four pots in each irrigation level on a single tray. We applied the corresponding volume of water to the tray every 8 days. We kept the plants from sowing to harvesting in a greenhouse within the botanic gardens of the University of Leipzig. In addition to sunlight, the greenhouse provided artificial light using HPI-T Plus 400W lamps (Philips). Lamps worked on a fixed schedule of 16/8 h light/darkness (6:00–22:00/22:00–6:00). The temperature within the greenhouse was not artificially regulated.

### 4.2. Irrigation/Tensiometers

We filled 3 L (Ø 19 cm) culture pots with a 1:1 (*v*/*v*) mix of soil (Floradur B pot clay medium-coarse, Floragard Vertriebs-GmbH) and sand (0/2 washed, Rösl Rohstoffe GmbH & Co. KG). We hydrated the soil in the pots by starting the four levels of irrigation (see Section 4.1). After the soil was hydrated, we inserted a full range tensiometer (Umwelt-Geräte-Technik GmbH) in the soil of each pot. We recorded the soil water tension every 15 min. We exported the tensiometer data as a .csv file and calculated average values for each irrigation. We recorded the air relative humidity and temperature in the greenhouse every 12 min using a sensor, model 224.401 (RAM GmbH Mess- und Regeltechnik, Herrsching). We plotted the average tensiometer readings, air relative humidity and temperature as scatter plots with the program Origin^®^ (v. 7SR1). We processed the original curves with Origin with the following parameters: smoothing: Savitsky-Golay, polynomial order=first, points to the left = 10, points to the right = 10.

### 4.3. Metabolite Extraction

On harvest day, the plants were ca. 11 weeks old. We collected one soil sample per plant for elemental analysis, see Section 4.6. Afterwards we thoroughly removed soil from roots by hand washing with plenty of tap water. We dissected plants into roots, stems, leaves, and when available, fruits and seeds. All sections were freeze-dried immediately afterwards. We registered the dry weight of every section. We ground the samples separately with a ball mill (MM400, Retsch GmbH) with the following settings: 30 Hz for 1 min (30 Hz for 30 s only for the case of leaves). We extracted 20 ± 2 mg of ground material in a 2 mL Eppendorf tube by adding 1 mL methanol/Millipore water (75:25 *v*/*v*). We sonicated the extracts for 15 min. We then spun-down the extracts at 15,000 g for 15 min. We transferred the supernatant to a new 2 mL Eppendorf tube and added 1 mL methanol/Millipore water (75:25 *v*/*v*) to the pellet for re-extraction. The extract was sonicated for 15 min. We spun-down the extracts at 15,000 g for 15 min. We mixed the two extracts and transferred 200 µL of the mixed extract to a new Eppendorf tube. We finally added 800 µL methanol/Millipore water (75:25 *v*/*v*) to obtain samples for Section 4.4 and Section 4.7.

### 4.4. LC-MS Metabolic Profiling and MS^2^

We separated and characterized metabolites by injecting 1 µL extract (Section 4.3) in a UPLC (Dionex 3000, Thermo Scientific) coupled with an MS-qToF (maXis impact HD, Bruker). The chromatograph was equipped with an autosampler with regulated temperature and extracts were kept at 4 °C pre- and post-injection. Compounds were separated in a column AcclaimTM RSLC 120 C18 (Thermo Scientific), external dimensions = 2.1 × 150 mm, particle size = 2.2 µm, pore size = 120 Å. The column operated at 40 °C. The binary phase consisted of solvent A: water/formic acid (0.1% v/v), solvent B: acetonitrile/formic acid (0.1% v/v). Gradient for solvent A: 0 min 95%, 2 min 95%, 15 min 60%, 20 min 5%, 22 min 5%, 25 min 95%, 30 min 95%. The flow was constant at 500 µL min^−1^. ESI source conditions were: end plate offset = 500 V, capillary = 3500 V, nebulizer = 2 bar, dry gas = 10 L min^−1^, dry temperature = 220 °C. Transfer line conditions were: funnels 1 and 2 = RF 300 Vpp, isCD energy = 0 eV, hexapole = 60 Vpp, quadrupole ion energy = 5 eV, low mass = 50 *m*/*z*, collision cell energy = 10 eV, collision RF 50 Vpp, transfer time = 60 µs, pre-pulse storage = 5 µs. The mass spectrometer operated with a mass range of 50-1400 *m*/*z* and a spectral acquisition rate of 3 Hz. Sodium formate clusters (10 mM) were used for calibrating the *m*/*z* values. The clusters mix consisted of 250 mL isopropanol, 1 mL formic acid, 5 mL 1M NaOH, and the final volume was adjusted to 500 mL. For selectively fragmenting specific ions (MS^2^), we applied to the conditions described above a scheduled precursor function in the program Bruker O-TOF modifying only the collision cell energy = 35 eV. Appendix A shows the list of target ions for fragmentation. We purchased standards of atropine and scopolamine (Sigma-Aldrich) as acetonitrile solutions with a concentration of 1 mg mL^-1^ and injected them separately as standards.

### 4.5. LC-MS Data Processing

Bruker .d files were converted to .cdf files with the program Compass DataAnalysis (Bruker). The .cdf files were transformed to .abf files using the program Abf Converter (Reyfics/RIKEN). The .abf files were processed with the program MS-DIAL (v. 3.52, RIKEN) with the following parameters: data collection: mass accuracy: MS^1^ tolerance = 0.01 Da, retention time begin = 0.7 min, retention time end = 30 min, mass range begin = 50 *m*/*z*, mass range end = 1400 *m*/*z*; peak detection parameters: minimum peak height = 1000 amplitude, mass slice width = 0.1 Da, smoothing method=linear weighted moving average, smoothing level = 3 scans, minimum peak width = 5 scans; alignment parameters settings: retention time tolerance = 0.05 min, MS^1^ tolerance = 0.015 Da. We normalized the alignments against the total ion chromatogram. We exported the normalized data matrix containing all the alignments as a .txt file (spectra type = centroid).

### 4.6. Interpretation of Mass Spectra and Generation of Compound Structure Hypotheses

We generated hypotheses for probable structures of compounds producing signals in the LC-MS analyses following a dual strategy described in the next two sub-sections.

#### 4.6.1. Processing of MS^2^ Mass Spectra with MetFamily

We predicted compound class membership using the web-based program MetFamily [34]. We imported the .abf files produced in Section 4.5 in MS-DIAL. We exported both the metabolite profile (raw data matrix) as a spreadsheet as well as the MS^2^ spectral library (representative spectra) as an .msp file from the .abf raw data files. We imported both the metabolite profile and the MS^2^ spectral library in MetFamily. We processed the data using the default parameters choosing a minimum spectrum intensity = 0. We applied a hierarchical cluster analysis using default parameters to resulting features clustered based on spectral similarity.

#### 4.6.2. Manual Interpretation of Mass Spectra and Ion Signal Relative Quantitation

We manually interpreted the full scan and MS^2^ mass spectra. When possible, we supported our structure predictions by comparison with mass and nuclear magnetic resonance spectra previously reported in scientific publications. When possible, we also compared our structure predictions against mass spectral data available in the databases MassBank (http://www.massbank.jp/), MassBank of North America (http://mona.fiehnlab.ucdavis.edu/) and the Human Metabolome Database (http://www.hmdb.ca/). For comparing the relative concentration of specific ions in different irrigations, we used the data contained in the .txt file exported in Section 4.5. After we manually grouped MS signals into hypothetical structures, we selected the most intense ion in each group and used its relative intensity value for comparison across irrigations or organs.

### 4.7. Quantification of Atropine and Scopolamine

We performed an absolute quantification of tropane alkaloids using an HPLC-MS-QqQ (EVOQTM, Bruker). We purchased standards of atropine and scopolamine (Sigma-Aldrich) as acetonitrile solutions with a concentration of 1 mg mL^−1^. We performed direct infusion of each standard following Bruker’s protocol for the determination of the optimal multiple reaction monitoring (MRM) conditions in positive mode acquisition. The optimal MRM conditions were as follows: (i) scopolamine: target mass at Q1 = 304.1 (*m*/*z*), collision energy at Q2 = 16 eV, dwell time at Q2 = 25 ms, quantifier mass at Q3 = 138.1 (*m*/*z*), scan time = 50 ms; (ii) atropine: target mass at Q1 = 290.1 (*m*/*z*), collision energy at Q2 = 21 eV, dwell time at Q2 = 25 ms, quantifier mass at Q3 = 124.1 (*m*/*z*), scan time = 50 ms. We then created calibration curves for each atropine and scopolamine. We diluted each standard with acetonitrile to obtain solutions with the following four concentrations: 1, 0.1, 0.01, and 0.001 ppm. We coupled the MRM method to the LC method used in Section 4.5. The temperature of the autosampler was kept constant at 4 °C. We used the following MRM scanning conditions: (i) scopolamine: retention time = 3.3 min, retention time window = 0.5 min; (ii) atropine: retention time = 4 min, retention time window = 1 min. Using the LC and MS conditions described above we analyzed 1 µL extract (Section 4.3) of each sample. We exported the raw chromatograms of standards for the calibration curve and the plant extracts chromatograms as .txt files. We plotted the raw data as a continuous line with the program Origin^®^ (v. 7SR1). We integrated the peaks areas for atropine and scopolamine using the Get Peak Information function in Origin.

### 4.8. Quantification of Elemental Nitrogen

Ground plants sections and soil samples (see Section 4.3) were air-dried in an electrical oven for 48 h. We dried plants sections at 70 °C and soil at 40 °C, respectively. Dried soil was sieved through a 2 mm mesh. We manually removed root residues or soil organisms. Plant sections and soil were ground with a ball mill for 30 s with 30 Hz (MM400, Retsch GmbH) in order to homogenize the sample and reduce the particle size to 100 µm or less. We dried for 2 h the plant and soil samples at 70 °C and 40 °C, respectively. We weighed the dried and ground samples into tin boats (Elementar Analysensysteme GmbH) with the following dimensions: 4 × 4 × 11 mm for plants and 6 × 6 × 12 mm for soil. For soil we added a standard of WO_3_ (Elementar Analysensysteme GmbH) in a 1:1 weight ratio sample/standard. We weighed the samples (10–15 mg for plants, 40–50 mg for soil) with a microbalance (Cubis MSA Sartorius AG), tightly folded the tin boat with the sample and compressed it with a press into a tight pellet. We weighed the final pellets and stored them in a desiccator until measurement. We quantified elemental nitrogen using a Vario MICRO cube analyzer (Elementar Analysensysteme GmbH). We analyzed the samples following the manufacturer’s protocol.

### 4.9. Additional Statistics

We performed one-way ANOVA and post-hoc Tukey Honest Significant Differences in *R* (v. 3.5.1) using the functions ‘aov’ and ‘TukeyHSD’, respectively.

## 5. Conclusions

The demand for tropane alkaloids in pharmaceutical applications is primarily supplied by cultivation of Solanaceae. Our results show that irrigation is an effective method for maximizing the production of specific tropane alkaloids in *Datura stramonium* cultivated under greenhouse conditions. We also present evidence for possible new enzymes and/or selective transportation mechanisms leading to a richer tropane alkaloid structural diversity. The discovery of stereospecificity in anisodamine isolated from different organs of *D*. *stramonium* opens the possibility of using this plant for the production of pharmaceuticals with potentially different bioactivities in an economically competitive fashion.

## Figures and Tables

**Figure 1 metabolites-09-00131-f001:**
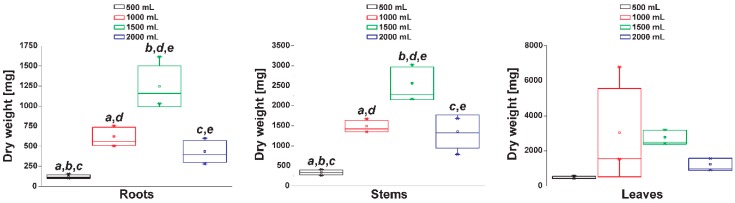
The dry weight of dissected *Datura stramonium* organs grown under different irrigations. Box plots show the standard deviation. Letters above bars represent significant pairwise differences according to Tukey test (ANOVA *p* ≤ 0.05).

**Figure 2 metabolites-09-00131-f002:**
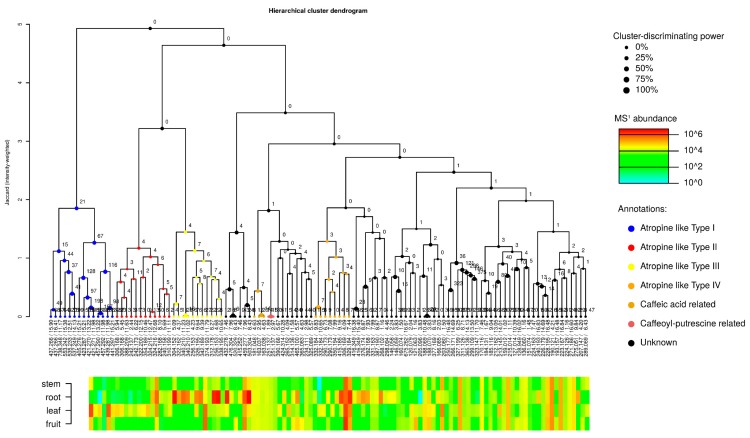
Prediction of compounds classes generated by MetFamily. Several MS^2^ signals are annotated as tropane alkaloids.

**Figure 3 metabolites-09-00131-f003:**
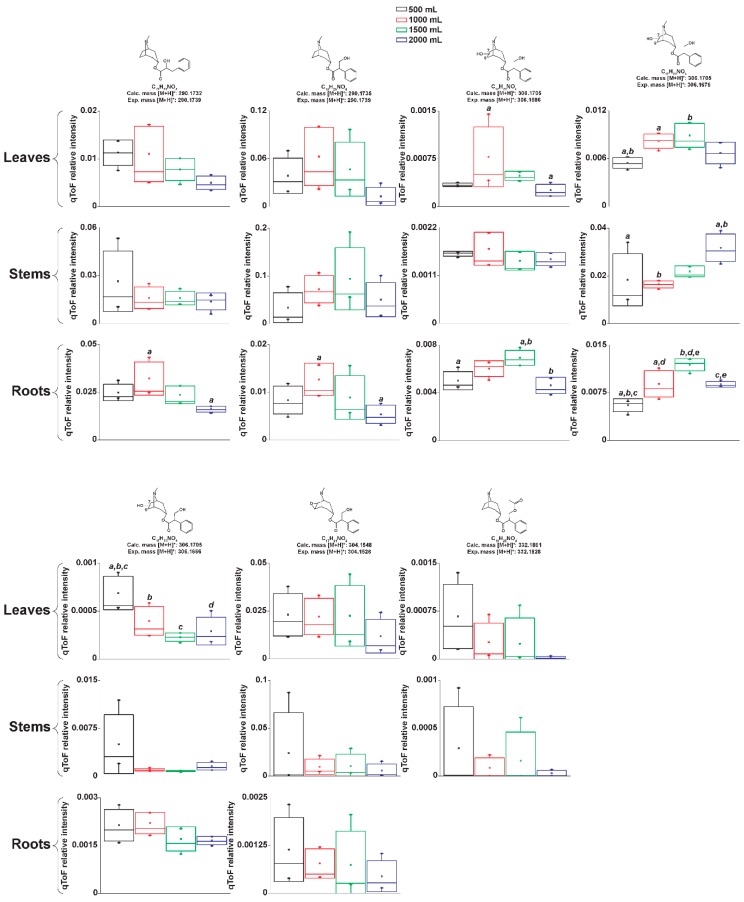
Aromatic tropane alkaloids and their relative concentrations in *Datura stramonium* grown under different irrigations. Atropine and scopolamine were identified by comparison against commercial standards. All other structures are predictions based on their mass spectra. Box plots show the standard deviation. Letters above bars represent significant pairwise differences according to Tukey test (ANOVA *p* ≤ 0.05).

**Figure 4 metabolites-09-00131-f004:**
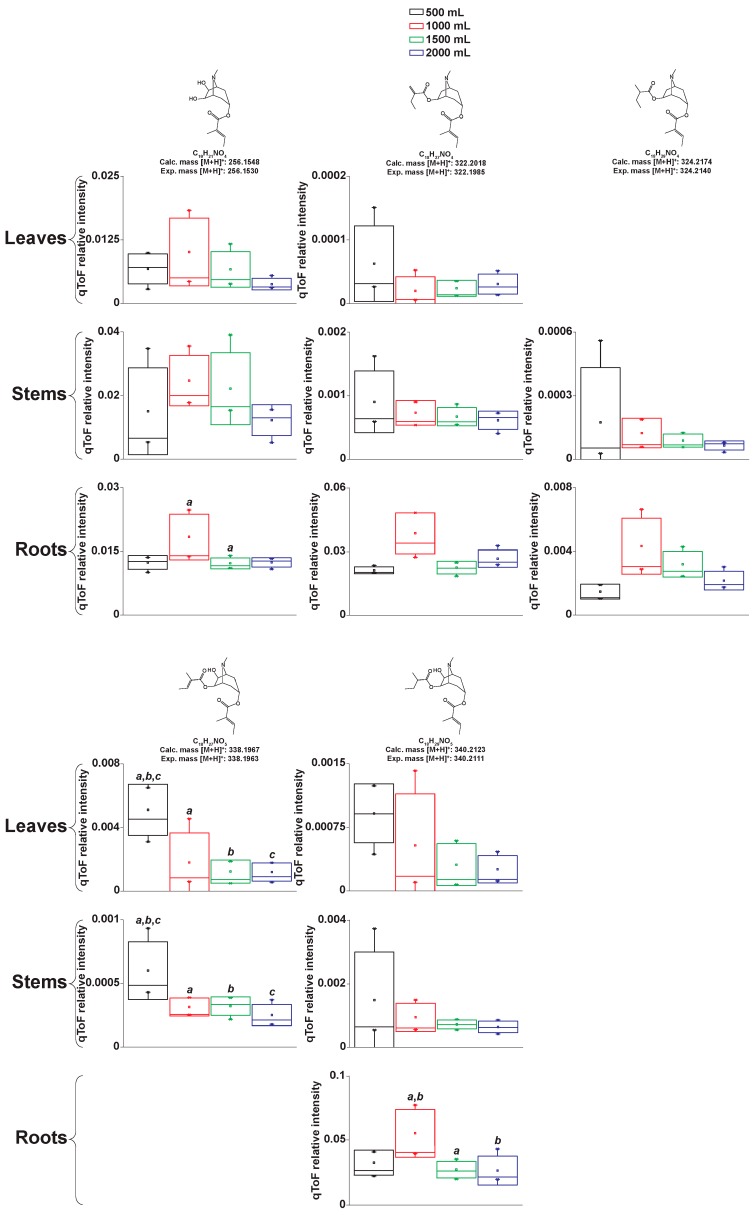
Aliphatic tropane alkaloids and their relative concentrations in *Datura stramonium* grown under different irrigations. Structures are predictions based on their mass spectra. Box plots show the standard deviation. Letters above bars represent significant pairwise differences according to Tukey test (ANOVA *p* ≤ 0.05).

**Figure 5 metabolites-09-00131-f005:**
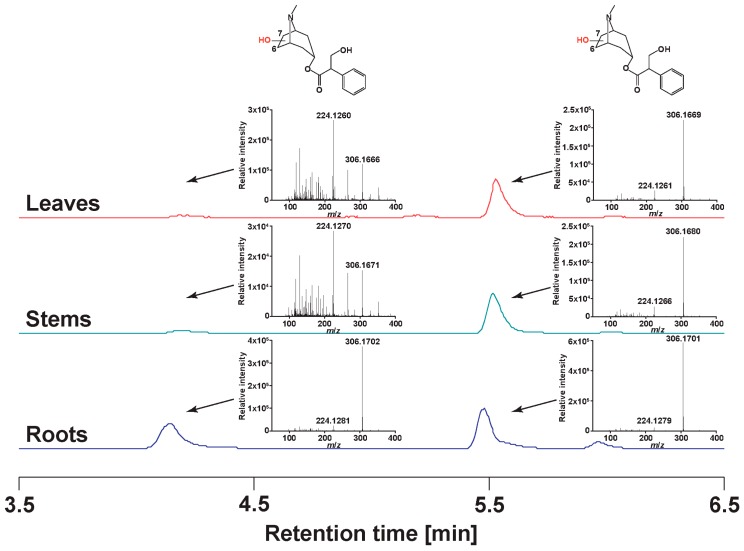
The relative concentration of putative anisodamine isoforms in *Datura stramonium*.

**Figure 6 metabolites-09-00131-f006:**
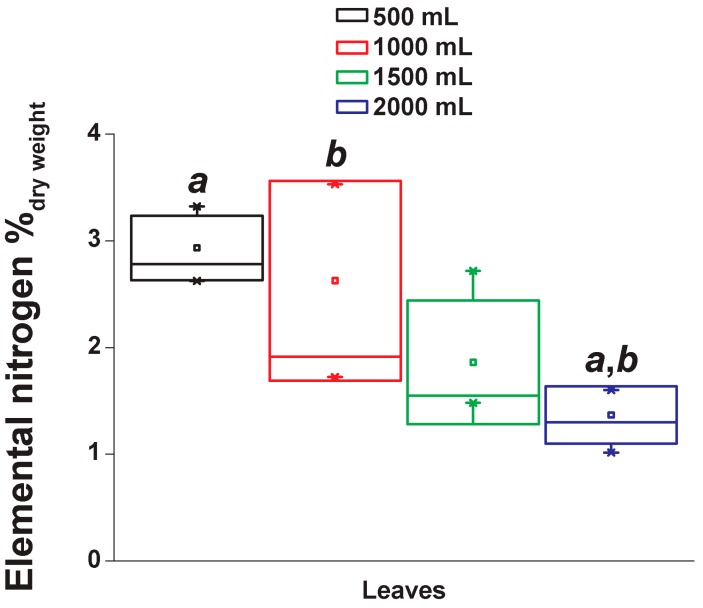
Leaf elemental nitrogen of *Datura stramonium* grown under different irrigations. Box plots show the standard deviation. Letters above bars represent significant pairwise differences according to Tukey test (ANOVA *p* ≤ 0.05).

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
