# Peer review of "Effects of Water Availability in the Soil on Tropane Alkaloid Production in Cultivated Datura stramonium"

_metabolites, 2019, doi:10.3390/metabo9070131_

Round 1

Reviewer 1 Report

The manuscript describes an intereting topic that water availability and tropane alkaloid production in cultivated Datura. Although it present a significant results from their experimental design, the authors should revise their manuscript in several ways before it is acceptable to publish.  The biggest concern would be no statistical tests on their main results: different concentrations of various tropane alkaloids.  Fig. 3 and Fig 4 shoud present ANOVA (analysis of Variance) results in addition to multiple comparison results.  Without those statistical evaluation, their conclusions can not be supported. 

Specific comments:

line 33: tropane alkaloids are found families other than Solanaceae.  For example, cocain is a tropane alkaloid from Erythroxylum coca in family Erythroxylaceae.  Also, family name is usually not italicized.

line 51: The genera name should not be abbreviated in the first place of a sentence. 

line 71: water tension concept might be explained more using water potential concept which is a standard theory on water relation in plants. For example, I would write like "water tension defined as sum of soil matrix potential and gravitational potential" 

line 73: "All four irrigation levels" has not been explained before so that I should go to material and method section to find what four irrigation levels are.  I would put a little detail such as "All four irrigation levels (500, 1000, 1500, and 2000 mL per 8 day)".

line 77: "Blox plots" should be a typo for "Box plots".

line 124: I don't understant why the authors present 2.5 Partial elemental composition section results.  They do not mention these results in discussion.  Anyway, in ecology, C:N ratio is an important variable indicating nitrogen content in plants.  The authors might search literature on C:N ratio and nutrient balance hypothesis if they would like to keep these results. For me, their results show increasing C:N ratios with more water availablity. 

line 134: I would like to see why irrgation affect tropane alkaloid production in the discussion section. Those explanation would provide a more strong mechanistic rationale for irrigation control to increase tropane alkaloid production.

Author Response

Prof. Dr. Peter Meikle

Editor-in-chief of Metabolites

Dear Prof. Dr. Meikle,

   I am addressing you this letter with the responses to the reviewers of the article:

Effects of water availability in the soil on tropane alkaloid

production in cultivated Datura stramonium

   On behalf of my co-authors, I thank the reviewers for their valuable observations. We consider that the modifications suggested by the reviewers made us improve the quality of our manuscript. Next, we will provide answers to every observation made by the reviewers.

Reviewer 1

The manuscript describes an interesting topic that water availability and tropane alkaloid production in cultivated Datura. Although it present a significant result from their experimental design, the authors should revise their manuscript in several ways before it is acceptable to publish. The biggest concern would be no statistical tests on their main results: different concentrations of various tropane alkaloids. Fig. 3 and Fig 4 should present ANOVA (analysis of Variance) results in addition to multiple comparison results. Without those statistical evaluations, their conclusions cannot be supported.

We have computed ANOVA and Tukey test for all the irrigation comparisons presented in the manuscript. We have modified the figures to show what results are statistical significant.

Line 33: tropane alkaloids are found families other than Solanaceae. For example, cocain is a tropane alkaloid from Erythroxylum coca in family Erythroxylaceae. Also, family name is usually not italicized.

We have clarified in the introduction that tropane alkaloids are produced by Solanaceae as well as Erythroxylaceae. We have edited the plant family descriptors throughout the text avoiding the use of italic fonts.

Line 51: The genera name should not be abbreviated in the first place of a sentence.

We have replaced the plant name with its full form.

Line 71: water tension concept might be explained more using water potential concept which is a standard theory on water relation in plants. For example, I would write like "water tension defined as sum of soil matrix potential and gravitational potential".

We have provided a more detailed description of the functioning of the tensiometers used in this study to quantify soil water availability.

Line 73: "All four irrigation levels" has not been explained before so that I should go to material and method section to find what four irrigation levels are. I would put a little detail such as "All four irrigation levels (500, 1000, 1500, and 2000 mL per 8 day)".

We have added the details referring to irrigation as suggested by the reviewer.

Line 77: "Blox plots" should be a typo for "Box plots".

We have corrected this typo in all the figure legends.

Line 124: I don't understand why the authors present 2.5 Partial elemental composition section results. They do not mention these results in discussion. Anyway, in ecology, C:N ratio is an important variable indicating nitrogen content in plants. The authors might search literature on C:N ratio and nutrient balance hypothesis if they would like to keep these results. For me, their results show increasing C:N ratios with more water availability.

We realized that presenting both Carbon and Nitrogen elemental percentages was confusing. Our objective was to investigate if there was any correlation between elemental nitrogen concentration and tropane alkaloids. We did not find such correlation. We explained this lack of correlation and the end of section 3.2 in our original submission. To make our point clear, in the revised version of our manuscript we removed the percentage of elemental carbon to emphasize that our message for the reader is that there is no correlation between the concentration of elemental nitrogen and specific tropane alkaloids.

Line 134: I would like to see why irrigation affect tropane alkaloid production in the discussion section. Those explanations would provide a more strong mechanistic rationale for irrigation control to increase tropane alkaloid production.

Our manuscript shows an exploratory investigation. We had no preconception of how soil water availability could influence, if any, tropane alkaloid composition in cultivated Datura stramonium. The most significant finding of our project is that different tropane alkaloids respond in different ways to irrigation. We do not have a mechanistic explanation for this compound specific variation. Since tropane alkaloids like atropine, scopolamine and anisodamine have been extensively studied as neurological agents and they show different potencies it is possible that in nature different tropane alkaloids produced by Datura stramonium affect differently different herbivores. Therefore, it is possible to speculate that natural selection selected Datura stramonium to produce specific tropane alkaloid profiles in response to soil water as this environmental factor might influence the herbivore community composition attacking these plants.

Reviewer 2 Report

see attached document

Author Response

Prof. Dr. Peter Meikle

Editor-in-chief of Metabolites

Dear Prof. Dr. Meikle,

   I am addressing you this letter with the responses to the reviewers of the article:

Effects of water availability in the soil on tropane alkaloid

production in cultivated Datura stramonium

   On behalf of my co-authors, I thank the reviewers for their valuable observations. We consider that the modifications suggested by the reviewers made us improve the quality of our manuscript. Next, we will provide answers to every observation made by the reviewers.

Reviewer 2

There is a complete and very well written manuscript describing a well-designed and well performed study on the effects of D. stramonium irrigation level on the production of atropine and related natural product metabolites. These natural products are of importance as starting materials for the product of certain drugs. This is an interesting and useful study and the manuscript should be published in the journal without any major revisions.

Suggested changes the authors might wish to consider:

1. The authors use of the terms “absolute concentration” and “relative concentration” could be made clearer. The reviewer assumes that “absolute concentration” refers to the concentration of the sample introduced into the LC-MS instrument, as determined by reference to standard compounds, and that “relative concentration” refers to the metabolite concentration in mass units per mass of dried plant material extracted (20 ± 2 mg). If this is the case, the authors might want to state it clearly.

   We use the terms “absolute concentration” and “relative concentration” depending on the technique we refer to in the text. We used two different LC-MS techniques to analyze the same extracts. For the absolute concentration determination of atropine and scopolamine we used a multiple reaction monitoring method based on a triple quadrupole mass spectrometer. These results are shown in the supplementary table S2. We use the term relative concentration to refer to the results obtained with the quadrupole/time-of-flight mass spectrometer that we used for the metabolomics profiling of the plant extracts.

2. Figure 1 clearly indicates that there is a strong effect of irrigation level on the total mass of plant material produced, with for instance the largest amount of dried root and stem material harvested at an irrigation level of 1500 mL, while the largest amount of dried leaf material is produced in either 1000 or1500 mL irrigation levels. However, Figures 3 and 4 indicate that that maximal amounts of various metabolites of interest are produced in different plant organs at different irrigation levels. For instance a maximum amount of atropine is clearly produced in leaves at 1000 mL irrigation level. Since to idea is to grow the plants under a selected irrigation level in order to maximize the production of these metabolites, it might be interesting for the authors to add to the manuscript a Total Yield Table. This added table could list the actual total mass yield of each metabolite per plant, that is summed over all three plant organs, at each irrigation level.

We multiplied the qToF signal intensities (relative concentration) times the sample weight to produce a table with total yields for the aromatic tropane alkaloids with pharmaceutical application. We added this table to the supplementary materials and indicated in the text, section 2.2.2.

3. As noted above the writing and organization of the manuscript is generally excellent. There are however a few places where the grammar and word choice could be improved.

Examples:

Line 210 change “11 week old” to “11 weeks old”.

We corrected the text accordingly.

Line 210 change “a soil sample” to “one soil sample”.

We corrected the text accordingly.

Line 310 change “Tropane alkaloid demand for” to “The demand for tropane alkaloids in”.

We corrected the text accordingly.

Line 310 change “primarily covered by” to “primarily supplied by”.

We corrected the text accordingly.